# Dietary Supplementation with Popped Amaranth Modulates the Gut Microbiota in Low Height-for-Age Children: A Nonrandomized Pilot Trial

**DOI:** 10.3390/foods12142760

**Published:** 2023-07-20

**Authors:** Oscar de Jesús Calva-Cruz, Cesaré Ovando-Vázquez, Antonio De León-Rodríguez, Fabiola Veana, Eduardo Espitia-Rangel, Samuel Treviño, Ana Paulina Barba-de la Rosa

**Affiliations:** 1Molecular Biology Division, Instituto Potosino de Investigación Científica y Tecnológica, San Luis Potosí 78216, Mexico; oscar.calva@ipicyt.edu.mx (O.d.J.C.-C.); aleonr@ipicyt.edu.mx (A.D.L.-R.); 2CONACYT-Centro Nacional de Supercómputo, Instituto Potosino de Investigación Científica y Tecnológica, San Luis Potosí 78216, Mexico; cesare.ovando@ipicyt.edu.mx; 3Tecnológico Nacional de México, Instituto Tecnológico de Ciudad Valles, Ciudad Valles 79010, Mexico; fabiola.veana@tecvalles.mx; 4Instituto Nacional de Investigaciones Forestales, Agrícolas y Pecuarias, Texcoco 56250, Mexico; espitia.eduardo@inifap.gob.mx; 5Facultad de Ciencias Químicas, Benemérita Universidad Autónoma de Puebla, Av. San Claudio S/N, Ciudad Universitaria, Puebla 72000, Mexico; samuel_trevino@hotmail.com

**Keywords:** 16S rRNA sequencing, diet, popped amaranth, gut microbiota, short-chain fatty acids

## Abstract

Amaranth has been recognized as a nutraceutical food because it contains high-quality proteins due to its adequate amino acid composition that covers the recommended requirements for children and adults. Since pre-Hispanic times, amaranth has been consumed as popped grain; the popping process improves its nutritive quality and improves its digestibility. Popped amaranth consumption has been associated with the recovery of malnourished children. However, there is no information on the impact that popped amaranth consumption has on gut microbiota composition. A non-randomized pilot trial was conducted to evaluate the changes in composition, structure, and function of the gut microbiota of stunted children who received four grams of popped amaranth daily for three months. Stool and serum were collected at the beginning and at the end of the trial. Short-chain fatty acids (SCFA) were quantified, and gut bacterial composition was analyzed by *16S rRNA* gene sequencing. Biometry and hematology results showed that children had no pathology other than low height-for-age. A decrease in the relative abundance of *Alistipes putredinis*, *Bacteroides coprocola*, and *Bacteroides stercoris* bacteria related to inflammation and colitis, and an increase in the relative abundance of *Akkermansia muciniphila* and *Streptococcus thermophiles* bacteria associated with health and longevity, was observed. The results demonstrate that popped amaranth is a nutritious food that helps to combat childhood malnutrition through gut microbiota modulation.

## 1. Introduction

Malnutrition is a term that encompasses many different manifestations of inadequate nutrition, including both undernutrition and obesity, which are characterized by an imbalance in energy intake and energy expenditure [1]. Undernutrition, defined as a deficiency of calories or a shortage of one or more essential nutrients, is a significant pediatric health problem and a pressing and overwhelming global health issue contributing to nearly half of all deaths in children under five years of age [2], mostly occurring in low- and middle-income countries where, at the same time, rates of childhood overweight and obesity are rising [3]. Assessment of a child’s health has been based on measurements of the body mass index (weight [kg]/height^2^ [m^2^]), but to recognize clinical signs of certain undernutrition problems, weight, length, height, or z-score values have been used to classify the nutritional status of children. In this sense, undernutrition is classified as stunting when there is low height-for-age, e.g., < −2 standard deviations (SD); wasting, when there is low weight-for-age (<−2 SD), and underweight, when there is low weight-for-height (<−2 SD). Malnutrition is also classified as moderate when it falls between −2 and −3 SD and severe when it falls below −3 SD [3]. According to the z-score, 165 million children under five years of age are stunted and 50 million children are wasting [4]. Being classified as stunted is a key risk factor for diminished survival, poor child and adult health, decreased learning capacity, and lost future productivity [5], and is considered as a multifactorial disease caused not only by inadequate protein/energy intake, but also other factors, such as poor sanitation, no access to clean drinking water, and inadequate psychosocial stimulation. All these factors cause disruption of the normal gut microbial ecosystem known as dysbiosis, which results in a predisposition to common infections, impaired immunity, and worsening malnutrition [6,7].

Growing evidence has indicated that dietary patterns and adequate food processing conditions are factors that affect food digestibility and functionality properties, and influence the composition, structure, and function of gut microbiota, and, therefore, human health [8,9]. It is also known that the quality and quantity of protein intake affects the metabolites produced by the gut microbiota [9,10,11,12]. Carbohydrates (resistant starch and dietary fiber) that escape digestion and absorption in the small intestine are also used by the microbiota and, through saccharolytic fermentation, lead to the generation of short-chain fatty acids (SCFA). SCFA, mainly acetate, propionate, and butyrate, are primary fermentation end products and represent the major flow of carbon from the diet through the microbiome to the host. SCFA have a positive influence on gut integrity and nutritional health by improving the energy yield, modulation of colonic pH, production of vitamins, and the stimulation of gut homeostasis, including anti-pathogenic activities [7]. 

Amaranth has been recognized as a nutraceutical food because it contains higher amounts of proteins (compared with traditional cereals, such as corn, wheat, and rice), but, most importantly, because of its high nutritive value due to its adequate amino acid composition, which covers the requirements recommended for children and adults [13]. Amaranth grains contain several encrypted peptides with antidiabetic, antihypertensive, and antioxidant functions [14]. They are rich in lipids containing several sterols, including tocopherols, and they are also a good source of squalene, a key metabolite in the sterol pathway. In relation to micronutrients, amaranth grains are a good source of minerals, including phosphorus, potassium, magnesium, calcium, iron, zinc, manganese, and selenium, and are also rich in vitamins (B2, B6, and E), niacin, and thiamine [13]. Since pre-Hispanic times, popped amaranth grain has been consumed, which was obtained by placing the grain on a clay pot heated with firewood, resulting in a pre-cooked food with a nutty flavor [15]. Currently, popped amaranth is obtained in a gas-fluidized bed with very short residence times, so popped amaranth could be considered as a minimally processed healthy food snack due to its high quality and quantity of proteins [16]. As for other plant-based food products, heat treatment processing does not alter amaranth grain properties but popping increases the digestibility of its nutrients [17], enhances its antioxidative properties [18,19], and reduces adverse antinutritional compounds, such as tannins, lectins, and trypsin inhibitors [17]. The consumption of popped amaranth grain has been associated with health benefits in humans, including recovery of severely malnourished children [17,20,21]. 

Recent reports have shown that the administration of quinoa protein and its hydrolysates was able to restore the gut microbiota in a mice model of colorectal cancer to similar to that of the control group [22]. Amaranth as quinoa is known as a super-grain because of its high nutritive quality; however, despite the extensive evidence that supports the beneficial effects on health of amaranth consumption, to date, there have been no studies that have analyzed the effects of popped amaranth consumption on gut microbiota composition. Therefore, the present study sought to utilize a nonrandomized pilot trial design to explore the effect of popped amaranth consumption on changes in the structure and abundance of the gut microbiota of children classified as low height-for-age (i.e., stunted children). The results obtained show that popped amaranth consumption helps to combat child malnutrition through gut microbiota modulation.

## 2. Materials and Methods

### 2.1. Recruitment of Participants

Children living in San Antonio Huichimal, Lima, and the Vista Hermosa rural area of Tenek, Ciudad Valles, San Luis Potosi, S.L.P., Mexico, were recruited for this research through screening based on inclusion and exclusion criteria. Sick children or those children with a reported disease were excluded. The selected children had not taken antibiotics in the last three months prior to the study. Body weight was measured using a digital weight scale with the infant wearing a light cloth and no shoes (accuracy: 0.1 kg). Meanwhile, body height was measured using a 2-meter-long microtoise without shoes (accuracy: 0.1 cm). Children were eligible for inclusion if they were aged between 6 and 7 years. They were grouped into two groups: a control group (Ctrl), including those that presented normal height-for-age with a mean of HAZ = −0.03 ± 0.5 (25 children), and a stunted or low height-for-age (HAZ < −2 SD) group (9 children).

To identify factors related to the children’s health status, a family survey was carried out. The survey included questions such as the type of home construction, access to basic services, type of food, type of child’s diet, type of house kitchen, and drinking water services.

### 2.2. Research Design

The study was approved by the ethics committees of the Health Services of San Luis Potosi (approval reference: SLP/006-2018), and DIF-Ciudad Valles, San Luis Potosí. This research was conducted following the applicable regulations and guidelines of the Helsinki Declaration, revised in 2000. Informed consent was signed by the participant children, as well as by the children’s parents/legal guardians. The trial was conducted from 13 March to 12 January 2021. Stunted children consumed four grams of popped amaranth daily for three months. Serum and stool samples from all participant children were obtained before and after the trial. Blood samples were used for chemistry and liver function analysis, while stool samples were used for SCFA analysis and gut microbiota composition. The research design is shown in Figure 1. 

### 2.3. Preparation of Amaranth Popped Grain 

Popped seeds were obtained by heating the seeds in an industrial hot air fluidized bed machine popper (Amaranta^R^, San Miguel de proyectos Agropecuarios, Hidalgo, México). The popped amaranth complied with the Official Mexican Standards (NOM-051-SCFI/SSA1-2010) for food and drinks for human consumption. The protein, fat, fiber, and ash contents were determined by standard methods [23]. Total carbohydrate was calculated by subtracting protein, fat, ash, and fiber from 100 [24].

### 2.4. Research Outcome

During the intervention period, the daily popped amaranth intake was supervised. Blood and stool samples were taken before and after the study. The primary outcomes were gut microbiota composition and SCFA profile, while a secondary outcomes was the hematic biochemical profile. 

### 2.5. Serum Collection, Hematic Cytometry, and Biochemical Profile Analysis

Blood samples (10 mL) were taken early in the morning under fasting conditions and following the established Official Mexican Norm (NOM-016S-SA3-2012). Children were requested to not take any food or water before blood collection. Blood was collected in BD Vacutainer^®^ Blood Collection tubes without anticoagulant. Samples were centrifuged at 400× *g* for 10 min; the serum was separated and frozen at −70 °C until analysis. Hematic cytometry was performed in an automatic system Kontrolab Model 5H+ (KontroLab Co., Ltd., Giudornia, Roma, Italy). Then, from each sample, a drop of blood was spread on a clear glass slide and stained with a May—Grünwald–Giemsa stain to differentiate the types of white blood cells in the sample. An aleotory count of 100 white blood cells in a bright-field microscope Axio Imager-A2m (Carl Zeiss Microscopy, White Plains, NY, USA) at 100× magnification was carried out. Serum concentrations of glucose, triglycerides, total cholesterol, urea, acid uric, creatinine, total protein, albumin, aspartate aminotransferase (AST), alanine aminotransferase (ALT), gamma-glutamyl transferase (GGT), alkaline phosphatase (ALKP), and lactate dehydrogenase (LDH) were determined by spectrophotometry using commercial kits (SPINREACT, Girona, Spain) and read on a MultiskanGO microplate spectrophotometer (ThermoFischer Scientific, Waltham, MA, USA). 

### 2.6. Stool Sample Collection 

For the collection of stool samples, a clean potty was placed in the toilet; immediately after defecation, stool samples were collected aseptically in a sterile stool container. The children were instructed to urinate first to avoid urine contamination; they were supported by their parents and trained personnel were in charge for stool collection. Samples were transported to the laboratory using ice packs, aliquoted, and immediately stored at −70 °C until further processing. 

#### 2.6.1. Measurement of Fecal Short-Chain Fatty Acids (SCFA) 

An aliquot of stool samples (250 mg) was homogenized in one mL of H_2_SO_4_ (0.5 mmol/L) and mixed at 1400 rpm for 3 min (Thermomixer, Eppendorf, Hamburg, Germany). The homogenized sample was incubated for 20 min in an ice-water bath and centrifuged at 4 °C and 4800× *g* for 15 min. The supernatant was recovered and subsequently centrifuged at 4 °C and 13,000× *g* for 15 min. This procedure was repeated two times for clarification. The sample was filtered through a 0.22 μm Millipore filter (Merck, Darmstadt, Germany) before injection into the chromatographic system. SCFAs were analyzed using an Agilent 1100 (Hewlett Packard, Santa Clara, CA, USA) and a Rezex ROA LC Column 150 × 7.8 mm (Phenomenex Inc., Torrance, CA, USA). The mobile phase was composed of H_2_SO_4_ (0.5 mmol/L). The column temperature was 60 °C, the flow rate was 0.5 mL/min, and measurement was performed using an RID-10A RI detector. Calibration curves were calculated from 0.18 to 1.8 mg/mL for acetic acid, 0.08 to 0.8 g/L for propionic acid, and 0.11 to 1.1 g/L for butyric acid. 

#### 2.6.2. DNA Extraction and Integrity Verification 

Stool samples (250 µg) were diluted in 1300 µL of saline solution (0.85%), removing coarse remains. A quantity of 600 µL of the decanted sample was taken and resuspended in a new tube. The suspension was centrifuged at 10,000× *g* for 10 min at 4 °C. The obtained pellet was resuspended in 1 mL of cold PBS and centrifuged at 700× *g*, 4 °C for 1 min. The supernatant was collected and centrifuged at 9000× *g* for 5 min at 4 °C. The resulting pellet was used for DNA extraction following the specifications of the DNeasy UltraClean Microbial Kit from QIAGEN (Hilden, Germany). DNA was quantified in a NanoDrop One (ThermoFischer Scientific, Waltham, MA, USA), and integrity was verified by visualization on a 1% agarose gel in a Tris-Borate-EDTA (TBE) buffer for 60 min at 70 volts. DNA extracts were stored at −80 °C until sequenced. 

### 2.7. 16S rRNA Gene Sequencing and Bioinformatics Analyses

For microbiome analysis, the V4–V5 region of the *16S ribosomal RNA* gene was amplified once using the universal bacterial primers 515FB: 5′-GTGYCAGCMGCCGCGGTAA-3′ and 926R: 5′-CCGYCAATTYMTTTRAGTTT-3′. A total of 51 Samples were sequenced on the Illumina MiSeq platform (Illumina, San Diego, CA, USA) using 300 + 300 bp paired-end according to the protocol described elsewhere [25]. Samples were sequenced at the Integrated Microbiome Resource (IMR) (Dalhousie University, Halifax, NS, Canada).

#### 2.7.1. Amplicon Sequence Variant Inference

The R package DADA2 v1.16.0 [26] was used to process the 16S sequencing data. We trimmed 20 nt to the right side of the forward fragments and 60 nt to the right side of the reverse fragments using the *filterAndTrim ()* function with default parameters. We used the *learnErrors()* function with default parameters to learn the error rates. We applied the *dada()* function to infer the sample composition using default parameters, the filtered fragments, and the calculated error rates. We merged the fragments with the *mergePairs()* function and removed the chimeras using the *removeBimeraDenovo()* function with default parameters. The taxonomy for each sequence was assigned using the *assignTaxonomy()* function and the silva_nr99_v138 database. The amplicon sequence variant (ASV) quantification and the phylogenetic assignment were obtained.

#### 2.7.2. Diversity Quantification and Functional Prediction

Alpha diversity, Shannon, inverse of Simpson, Fisher, Chao1, and ACE indices were calculated with the *diversity()* function from the R package vegan v2.5.6 [27]. To determine the structural variation of microbial communities, beta diversity and NMD plots were generated with custom R scripts and using the BrayCurtis dissimilarity calculated with the *ordinate()* function from the vegan package.

The functional prediction of the 16S sequencing data was performed with Tax4Fun2 [28]. The *RunRefBlast()* and *makeFunctionalPrediction()* functions were used with default parameters to predict the functional profiles of the ASV quantification results. The reference used in both steps *runRefBlast* and *makeFunctionalPrediction* was RF99NR. The pathway predictions table (pathway scores per library) was used to perform a sparse partial least squares discriminant analysis (sPLS-DA). The function *spls-da()* from the R package mixOmics with default parameters and *ncomp = 2* was used [29]. The plots to represent the results of the discriminative analysis (DA) were generated with custom R scripts. 

Using significant taxa and routes, taxa-pathways networks were created (permutation test spls-da, 1000 permutations, *p*-value < 0.1). The Spearman correlation between two nodes (taxa or pathways) was used to identify their connection using the igraph R package [30]. Only correlations greater than the 30th percentile were employed to create the networks.

### 2.8. Statistical Analyses

The statistical analysis of results was performed on Prism 8.0 for Mac (GraphPad Software, San Diego, CA, USA). The normal distribution was tested by D’Agostino–Pearson and Shapiro–Wilk tests. One-way ANOVA was performed for data with a normal distribution, followed by Bonferroni’s post hoc test. A Kruskall–Wallis test, followed by Dunn’s post hoc test, was carried out if data were not normally distributed. A *p*-value less than 0.05 (*p* < 0.05) was considered as statistically significant. 

## 3. Results and Discussion 

### 3.1. Popped Amaranth as Minimally Processed Food 

Popping is a traditional process that has been carried out in Mexico since pre-Hispanic times. At that time popping was obtained by heating the amaranth grains for 15–25 s in covered clay pots at 175 to 195 °C, which resulted in grain expansion [31,32]. Currently, popped amaranth is obtained in an industrial hot air fluidized bed machine at 300 to 330 °C with 9 s of residence; hence, the popped amaranth could be considered a minimally processed food. Under these conditions, popped amaranth showed high protein content (15.8%), fat (6.7%), fiber (3.8%), and ashes (2.2%) (Appendix A).

### 3.2. Children’s Selection and Sociodemographic Living Conditions Questionaries

Children showing undernutrition and classified as stunted based on the height-for-age parameters [3] were selected and named as the undernutrition group (UN, *n* = 9). The UN group daily consumed four grams of popped amaranth and were named the UNA group, but during the trial and COVID-19 pandemic situation, two children abandoned the study (UNA, *n* = 7). A control group (Ctrl, *n* = 21) represented by children of normal height-for-age living in the same conditions as the UN group was also selected.

The main results of the sociodemographic questionaries showed that factors associated with children’s health status were related to living conditions, such as the use of latrines (not properly a bathroom), firewood used as fuel in kitchens, and the dormitory was the same place as the kitchen. The age of the mother at the birth of the child ranged from 20–24 years and most had high school as the highest level of studies. Although, in general, all the children had access to basic food (beans, rice, tortillas), undernourished children showed low appetite and normally ate only once a day, while the control children ate three times a day. Some other problems, such as parasitic infections, were detected in the children [33].

### 3.3. Children’s Serum Biochemical Analysis

The hematic cytometry results for the participating children are shown in the Appendix A. The baseline values for participants reflected that they did not have any other pathology than low height-for-age in the UN group. However, although all parameters fell within biological limits, it was observed that hematocrit and mean globular volume (MCV) values had a tendency to increase in the UNA children, compared with the control group. Low values of MCV are related to protein-energy malnutrition or deficiency of iron and folate [34]. Amaranth is a rich source of protein and contains high amounts of minerals, such as iron, calcium, and magnesium [13]. This could be one of the reasons for the observed trend of an increase in these values and reported child health improvement. 

The serum biochemical profile did not show any pathology. The renal function, tested by creatinine, uric acid, and urea, was normal in all participants. The biomarkers for liver function (ALT, TGO) showed normal levels. No significant changes were observed in cholesterol, triglycerides, and high-density lipoprotein (HDL) amongst the groups (Appendix A). 

### 3.4. Modulation of Gut Microbiota in the Undernutrition Children’s Group after Amaranth Consumption

Sequencing of the *16S ribosomal ARN* gene (*16S rRNA*) has generated great interest in the study of bacterial communities. Amplicon sequencing variants (ASVs) have been proposed as an alternative to operational taxonomic units (OTUs) to analyze microbial communities. The use of ASVs has grown in popularity because they reflect a more refined level of taxonomy; however, the use of ASVs must be undertaken carefully in order to avoid the risk of dividing a single bacterial genome into separate clusters [35]. Once the sequences were refined, a total of 90,904 ASVs were obtained for the microbiome of the control children, 105,312 ASVs for the underweight children, and 91,526 for the underweight children after amaranth consumption.

The alpha diversity, which reflects the gut microbial richness was assessed based on the observed species and the Shannon index [36]. Although no significant differences amongst groups were observed, a tendency toward lower values was observed in the UN group; however, after amaranth consumption, the UNA group showed a tendency to increase to similar values to those of the Ctrl group (Figure 2A,B). Simpson’s inverse index, which indicates the diversity or dominance of species in the sample, also showed a tendency to decrease in the UN group, which suggests that the bacterial community was smaller in this group. However, this value also increased after amaranth consumption for the UNA group (Figure 2C), and the same tendency was observed for Chao1 (Figure 2D), and for the Fisher, ACE, and Simpson values (Appendix A).

Previous reports have indicated that children with undernutrition show lower gut microbiota diversity [12,37,38], which agrees with the observed results in the present work. It was also reported that the Shannon index showed a tendency to decrease in malnourished children with protein malnutrition (Kwashiorkor), although no significant values were reported [39]. It is known that results in humans are sometimes not statistically significant because of the intrinsic variation among people, but they are clinically relevant as they reflect a clear trend [40]. The beta diversity reflects the differences in the gut microbial composition of different groups and is represented by non-metric multidimensional scaling (NMDS) or PCoA, based on the Bray–Curtis dissimilarity. As shown in Figure 3, PCoA analysis showed a difference in the gut microbiota between the Ctrl and UN groups; however, after amaranth consumption, the UNA group tended to be more like the Ctrl group. 

The major phylum in the gut microbiota of children in the Ctrl group was the Actinobacteriota (28.8%), followed by the Bacteroidota and the Firmicutes (40.2%), Proteobacteria (12.8%), and Verrucomicrobiota (16.8%). It was observed that the UN group showed a significant increase in Actinobacteriota (50.9%) with loss of the Verrucomicrobiota phylum, which agrees with data reported by Kamil et al. [41]. After the trial, the UNA group showed a significant decrease in the abundance of Actinobacteria (20%) with an increase in Bacteriodotes and Firmicutes. Interestingly, Verrumicrobiota were recovered after amaranth consumption with similar values as the control group (13.4%) (Figure 4A). It has been reported that the Proteobacteria is the phylum that contributes to dysbiosis [42]; however, in this work, this phylum did not show significant changes in abundance amongst groups (12.8 to 15.4%) (Figure 4A). 

It is considered that the Firmicutes and Bacteroidetes ratio (F/B) is important for determining health status, and together represent the dominant phyla in the gut microbiota. Change in the F/B ratio has been associated with several diseases; however, it is important to be aware that this ratio could be affected by the change in abundance of all the other phyla, as observed in the present results. Our data show that, after amaranth consumption, these two phyla reached up to 51.6% in the balance abundance of F/B (Figure 4B). It has been reported that a diet high in sugar and low in fiber in children with undernutrition might cause a high F/B ratio [43]. An abundance of Firmicutes has been associated with the modulation of calorie absorption effectiveness by increasing the number of lipid droplets [44], and a high F/B ratio has been associated with the incidence of obesity [45]. Lower abundance of Firmicutes with higher abundance of Bacteroidetes has been associated with inflammatory bowel disease. It is inferred that the balance of these two phyla is important for maintaining normal gut homeostasis [46].

At the family level, a high abundance of Eubacteria was observed in the UN group with a significant decrease after amaranth consumption (24.8% and 11.1%). By contrast, the Akkermansiaceae family was found at very low abundance (3.3%) in the UN group; however, this family increased after amaranth consumption (9.1%) to an even higher abundance than the control group (1.0%). The Alcaligenaceae family was significantly increased in abundance after amaranth consumption (11.7%), and the Bacteroidaceae family, that was almost depleted in the UN group (2.6%), increased after amaranth consumption to similar values to the control group (5.9 and 7.6%, respectively). The Christensenellaceae family was also found in high abundance after amaranth consumption (7.8%), while the Coriobacteriaceae decreased from 7.2% in the undernourished group to up to 2.0% after amaranth consumption (Figure 4C).

### 3.5. Modulation of Gut Microbiota at Family-Genus Level was Observed in Undernourished Children after Amaranth Consumption

As described above, although the F/B ratio provides an estimation of a person’s health, care should be taken because this ratio could be affected by changes in the other phyla, as well as the type of genus that changes in each phylum. After amaranth consumption most of the species’ changes in relative abundance were observed in the Firmicutes phylum.

#### 3.5.1. Changes in Firmicutes Phylum at Genus/Species Level 

It has been reported that species of the Lachnospiraceae family belonging to the Eubacteriales order occur in over-abundance in undernourished children, with some of the members of this family serving as metabolic regulators in undernourished individuals [43]. Our results also showed high abundance of this family in the undernutrition group (Figure 4), and changes in the relative abundance of *Eubacterium hallii*, *Ruminococcus gauvreauii*, *Blautia faecis*, *Blautia obeum*, CAG-56, *Fusicanilobacter, Roseburia hominis*, and *Roseburia intestinalis*, amongst others, were observed after amaranth consumption (Figure 5). *E. hallii* was observed to decrease in the undernutrition group, but after amaranth consumption there was a relative increase. *E. hallii* is an anaerobic bacterium that belongs to the group of butyrate producers [47], and its administration in mice with diabetes changed the function of the intestinal microbiome, improving the metabolic phenotype [48]. 

At the genus level, *Blautia* has attracted particular interest since its establishment in the gastrointestinal tract has been related to the alleviation of inflammatory and metabolic diseases as well as to antimicrobial activity. This bacterium has also been reported to play a role in the crosstalk with other intestinal microorganisms [49]. *Blautia* actively participates in intestinal immunomodulation in human health [38,48], improves serum HDL-C, and a relative increase in its abundance was reported after a diet intervention for obesity [50]. Although *B. obeum* has been related to obesity [46], its abundance in undernourished children after amaranth consumption was at the same level as in control children (Figure 5). 

*Roseburia hominis*, and *Roseburia intestinalis* were identified; both bacteria showed low changes in their relative abundances after amaranth consumption, while the first decreased, the second increased. The *Roseburia* genus is described as comprising beneficial organisms as they are butyrate-producing bacteria and both species have been found to be depleted in the stools of undernourished children [37]. *R. hominis* was positively related to colonic mucosal melatonin levels, a pineal hormone that can maintain circadian rhythms and regulate immune, antioxidant, and anti-inflammatory functions, alleviating the symptoms of digestive disorders, such as irritable bowel syndrome and ulcerative colitis [51]. It is also known that *R. intestinales* prevents inflammation and maintains energy homeostasis [52].

Within the Erysipelotrichaceae family, *Holdemanella*, was found to increase relatively after amaranth consumption (Figure 5). In malnutrition, it has been shown that *Holdomanella* was depleted [53]. *Holdomanella* has been isolated from the feces of healthy metabolic volunteers and has been linked to the amelioration of hyperglycemia, improvement in oral glucose tolerance, and restoration of gluconeogenesis and insulin signaling in the liver of obese mice [54]. 

The Streptococcaceae family was represented by a relative increase in the abundance of *S. salivarus* and *S. thermophilus* after amaranth consumption (Figure 5). *S. thermophilus* is a well-known probiotic and is commonly found in yogurts [55]. *S. salivarius* is a commensal in humans and is phylogenetically close to *S. thermophilus* [56]. Within the Oscillospiraceae family, *Subdoligranulum* showed a high relative abundance after amaranth consumption. *Subdoligranulum* is a butyrate-producing bacterium with high potential as a probiotic to handle metabolic diseases. It also affects the activity of acetyl-CoA acetyltransferase, acetyl/propionyl-CoA carboxylase, and butanol dehydrogenase, which contribute to butyric acid production [57]. Butyrate has several beneficial properties that are essential to maintain gastrointestinal health. Therefore, the butyrate-producing bacteria represent a niche-specific next-generation of probiotics with *Butyricicoccus* species being one of the potential bacteria [58]. *Butyricicoccus* have also been shown to have potential as a therapeutic target for food allergy [59]. *Butyricicoccus* showed an increase in relative abundance in undernourished children after amaranth consumption. 

#### 3.5.2. Changes in the Bacteroidetes Phylum at Genus/Species Level 

Within the Bacteroidota phylum, a relative increase in *Bacteroides coprocola* and *Bacteroides stercoris* was observed in the undernutrition group; however, after amaranth consumption their relative abundance decreased as well as that of *Alistipes putredinis* (Figure 5). Different strains of *B. coprocola* have been reported, some are related to type 2 diabetes, some have been identified in patients with hypertension, and some others have been linked to patients with attention-deficit/hyperactivity disorder [60]; thus, it is suggested to be cautious about its presence or absence [61,62]. Although *B. stercoris* is an SCFA-producing bacterium, some studies have linked its presence to ulcerative colitis as well as to that of *Alistipes putredinis* [63].

A decrease in the relative abundance of *B. fragilis*, *B. ovatus*, and *B. plebeius* was observed after amaranth consumption when compared with the control group (Figure 5). *B. fragilis* is a commensal organism but has also been reported as an opportunistic pathogen in clinical infections, and as being responsible for a variety of diseases involving disruption of the intestinal barrier [64]. Other studies have reported that in mice the non-toxigenic *B. fragilis* strain may mediate beneficial interactions with the host by directing the immune response and suppressing intestinal inflammation [65]; thus, a deep characterization should be carried out to identify the *B. fragilis* species detected in undernourished children. Some of the *B. ovatus* strains have been related to the induction of gut IgA and maintenance of tissue homeostasis [66]. *B. plebeius* was found to be a dominant bacterium in children with multisystemic inflammatory syndrome [67]. Only *B. intestinalis* showed a high relative abundance after amaranth consumption (Figure 5). This bacterium has been identified as a type of polyamine-producing bacteria, such as putrescine, spermidine, and spermine, all of which are organic cations required for animal cell growth and differentiation. They are also involved in various steps of DNA, RNA, and protein synthesis [68].

#### 3.5.3. Changes in the Verrucomicrobiota Phylum 

Amaranth consumption induced a relative increase in abundance of *Akkermansia muciniphila* (Figure 5). *A. muciniphila*, is a colonic mucin-degrading bacterium that has been linked to intestinal health and immune signaling and has been associated with reduction in the incidence of diabetes when administered as a probiotic [69]. It was reported that *A. muciniphila* improves metabolic disorders in dietary obese mice and was found to increase after resistant starch intake. It has been proposed as a new probiotic with the capacity to promote healthy longevity [70]. *A. muciniphila* can metabolize the mucosal layer as a source of carbon and nitrogen to produce acetate and propionate, playing an important role in maintaining the gut barrier, especially when enteral nutrition intake is low, such as during long-term fasting and in malnutrition [71]. 

The diversity and richness of the mouse microbiota changed when mice were fed fermented soybeans, resulting in enhancement of *Lactobacillus*, *Butyricicoccus*, members of the Lachnospiraceae family, and *A. muciniphila* [72]. Our results showed that these species were also found in increased abundance in undernourished children after amaranth consumption.

### 3.6. Amaranth Consumption Promotes Gut Microbiota-Dependent Metabolites 

Popped amaranth grains, in addition to high-quality proteins, are rich in carbohydrates that escape digestion (resistant starch), as well as dietary fiber from testa [13]. The gut microbiota is predominantly involved in the fermentation of indigestible carbohydrates into SCFA, which exert multiple effects on energy homeostasis and are crucial for intestinal health. The most abundant SCFA are acetate, butyrate, and propionate, comprising > 95% of the SCFA content [73]. Investigations have shown that SCFA concentrations in children with severe acute and moderate undernutrition are low, especially propionic and butyric acid, and these SCFA increase during recovery along with the gut microbiota number. In this study, it was observed that, after amaranth consumption, the levels of acetic, butyric, and propionic acids showed a tendency to increase, although only propionic acid was significant as well as the sum of these three SCFAs (Figure 6). *E. hallii*, *Ruminococcus*, *Blautia*, *R. hominis*, and *Butyricicoccus*, known as butyrate-producer species [40], were all observed in increased relative abundance after amaranth consumption. *A. muciniphila* and *Subdoligranulum* sp. have been described as acetate- and propionate-producer bacteria, respectively [57,71]—both were also found in increased abundance after amaranth consumption. Altogether, these data suggest that SFCA levels are restored after amaranth consumption, correlating with an increased abundance of SCFA-producer bacteria. 

The effect of proteins in the gut on the microbiota metabolism is less well understood since the high variability of proteins due to their amino acid composition and the presence or absence of essential amino acids induces differences in protein digestion by the microbiota generating a great diversity of metabolites, many of which could be detrimental to host health [74]. Recent work using quinoa protein isolates showed that they had a positive effect on the modulation of mice gut microbiota [22], similar to the effect observed in the present work; however, previous works using mice as models, when fed soy proteins showed that, in addition to a tendency to fat accumulation, the effect on the modulation of the gut microbiota was not the same compared to feeding of amaranth proteins [75]. Further work should be carried out in order to understand the effect of dietary protein on the protein–gut–host–health axis.

### 3.7. Functional Prediction of Bacterial Taxa

The metabolic pathway predictions of gut microbiota composition in the undernutrition group compared to the control group (UN/Ctrl) (Figure 7) showed increased starch and sucrose metabolism, which decreased after amaranth consumption (UNA/Ctrl). The biosynthesis of secondary metabolism was increased after amaranth consumption, as well as butanoate, pyruvate, and propanoate metabolism, which are the routes to produce butyric and propionic acid, associated with the tendency to increase of these metabolites after amaranth consumption (Figure 7).

Purine and amino sugar and nucleotide metabolism, as well as the pathways related to vitamin metabolism, such as biotin (vitamin H), nicotinate and nicotinamide (B3), and riboflavin (B2), showed a tendency to increase after amaranth consumption. Increase in metabolic pathways related to sphingolipid and arachidonic acid metabolism was observed after amaranth consumption. The sphingolipids are considered to be a class of bioactive lipids that play key roles in the regulation of several cellular processes, while arachidonic acid forms part of cell membranes and is important for human health and tissue homeostasis. A correlation between Ruminococcaceae UCG 009 and arachidonic acid has been reported [76]. Several *Ruminocococcus* species were detected here after amaranth consumption (Figure 5). 

A network of correlations with metabolic activities was obtained and as shown in Figure 8, *Akkermansia* appeared to be an important species in controlling *Ruminococcus*, *Bacteroides*, and *Blautia*. There was also a positive relation with *Klebsiella* and *E. hallii*. *Akkermansia* also showed a positive correlation with species from the Oscillospiracea family, such as UCG-005, 003, and 002. The UCG strains regulate Christensenellaceae R-7 and *Butyrivibrio*, which is an important species in lipid metabolism. These groups of bacteria have a role in the regulation of nucleotide and terpenoid metabolism, transport, signal transduction, and replication and repair.

## 4. Conclusions

Popped amaranth has been promoted as a food source with high potential to combat protein-energy malnutrition. Our results show that popped amaranth intervention induces an increased relative abundance of *A. muciniphila*, a bacterium related to intestinal health and host longevity, as well as *Subdoligranulum*, which is considered to be a new class of probiotics. We also observed decrease in the relative abundance of *B. coprocola* and *B. stercoris*, both related to inflammation and colitis. In summary, the present work highlights the potential uses of popped amaranth as a source of plant-based proteins, which requires minimum processing to achieve its biological functions in health. However, eradication of malnutrition does not occur only through dietary changes, it is also important to improve the infrastructural conditions and education in rural areas. Further work in a larger randomized controlled trial will be important to confirm the beneficial effects of popped amaranth consumption.

## Figures and Tables

**Figure 1 foods-12-02760-f001:**
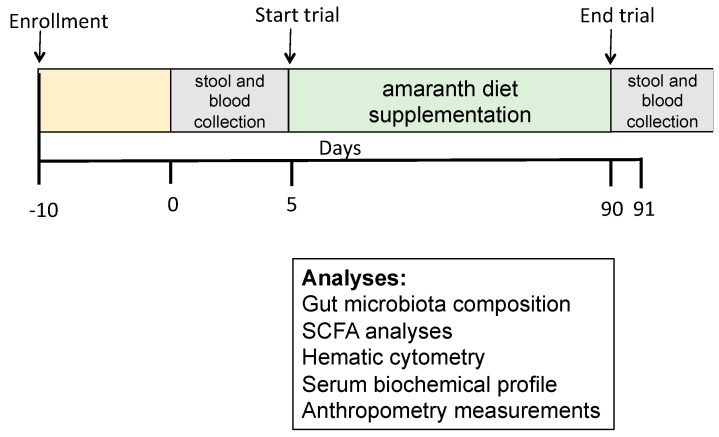
Study design to analyze the effect of supplementing the normal diet with amaranth on gut microbiota of children between 6 to 7 years old with moderate acute undernutrition living in rural areas.

**Figure 2 foods-12-02760-f002:**
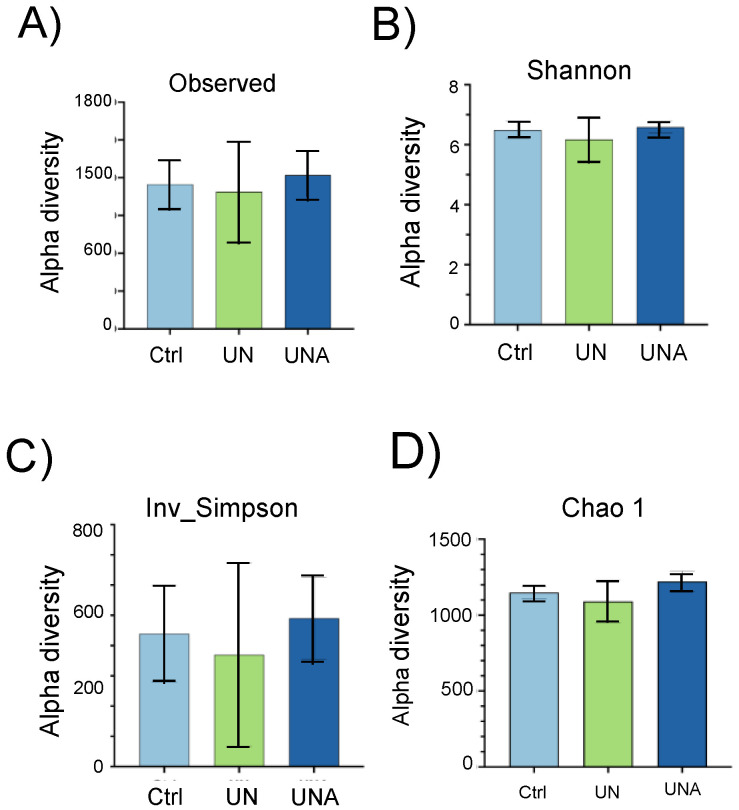
The effect of amaranth consumption on the gut microbiota composition of children living in rural areas. Alpha-diversity expressed as (**A**) observed richness, (**B**) Shannon index, (**C**) Inv-Simpson, and (**D**) Chao 1 indexes. Boxes express the IQR (interquartile range), bars indicate the minimum and maximum values. A Kruskall–Wallis test and Dunn’s post hoc analysis (*p* < 0.05) were performed. No significant differences were found amongst the groups and different indexes. Ctrl = control group of children with normal height-for-age; UN = undernutrition group with low height-for-age; UNA = UN group after three months of amaranth consumption.

**Figure 3 foods-12-02760-f003:**
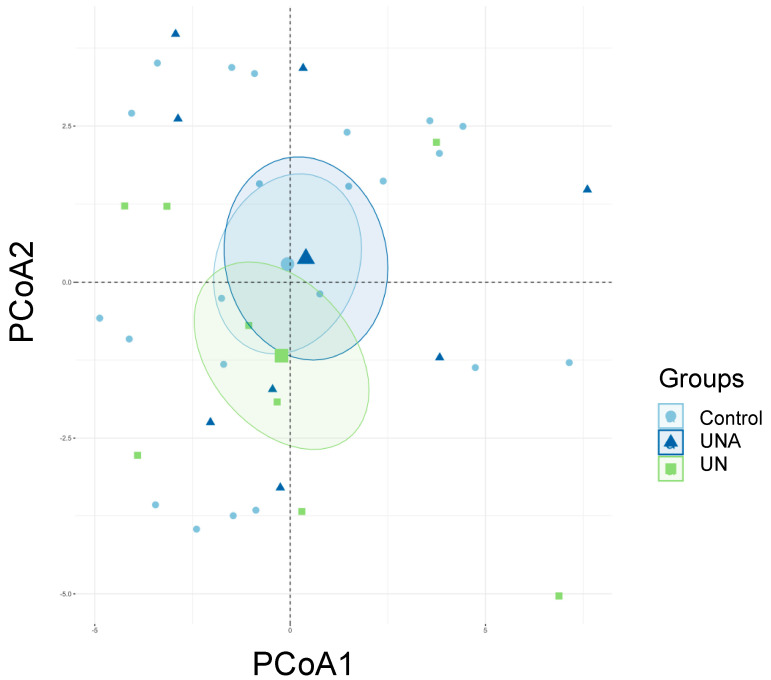
The effect of amaranth consumption on the gut microbiota composition of children living in rural areas. Beta-diversity measured using principal coordinate analysis (PCoA) based on the Bray Curtis dissimilarity (*p* < 0.05). Ctrl = control group of children with normal height-for-age; UN = undernutrition group with low height-for-age; UNA = UN group after three months of amaranth consumption.

**Figure 4 foods-12-02760-f004:**
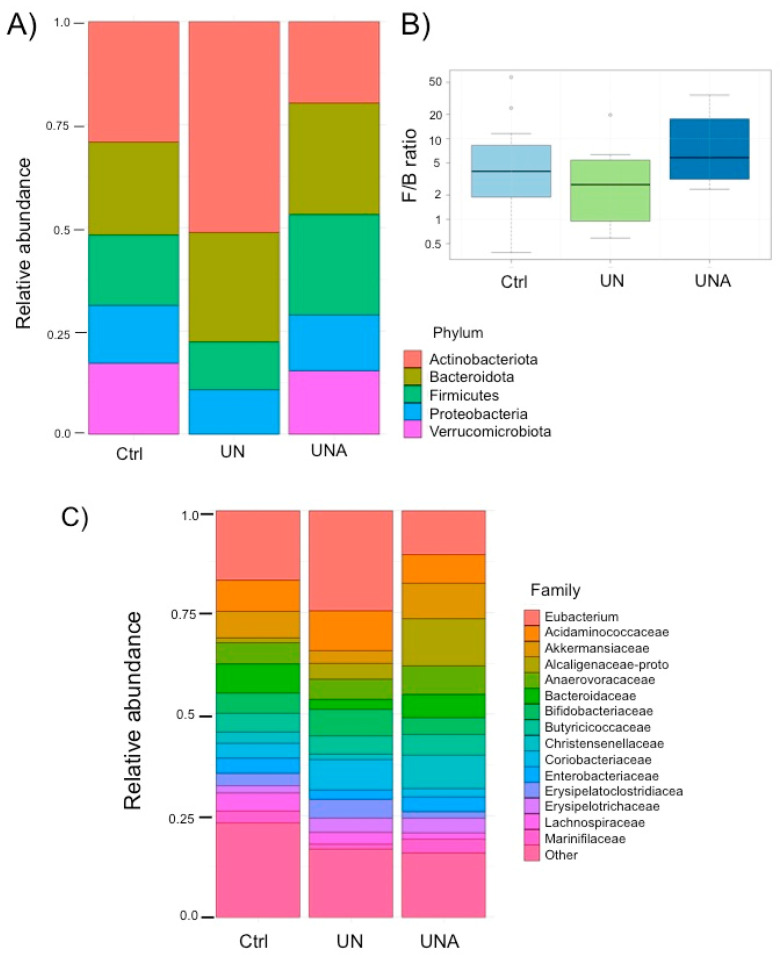
(**A**) Relative abundance in gut microbiota at phylum level; (**B**) Ratio of Firmicutes to Bacteroidetes (B/F). Significant differences were calculated using the non-parametric Wilcoxon test and non-significant differences were detected. (**C**) Relative abundance in gut microbiota at family level. Ctrl = control group of children with normal height-for-age; UN = undernutrition group with low height-for-age; UNA = undernutrition group after three months of amaranth consumption.

**Figure 5 foods-12-02760-f005:**
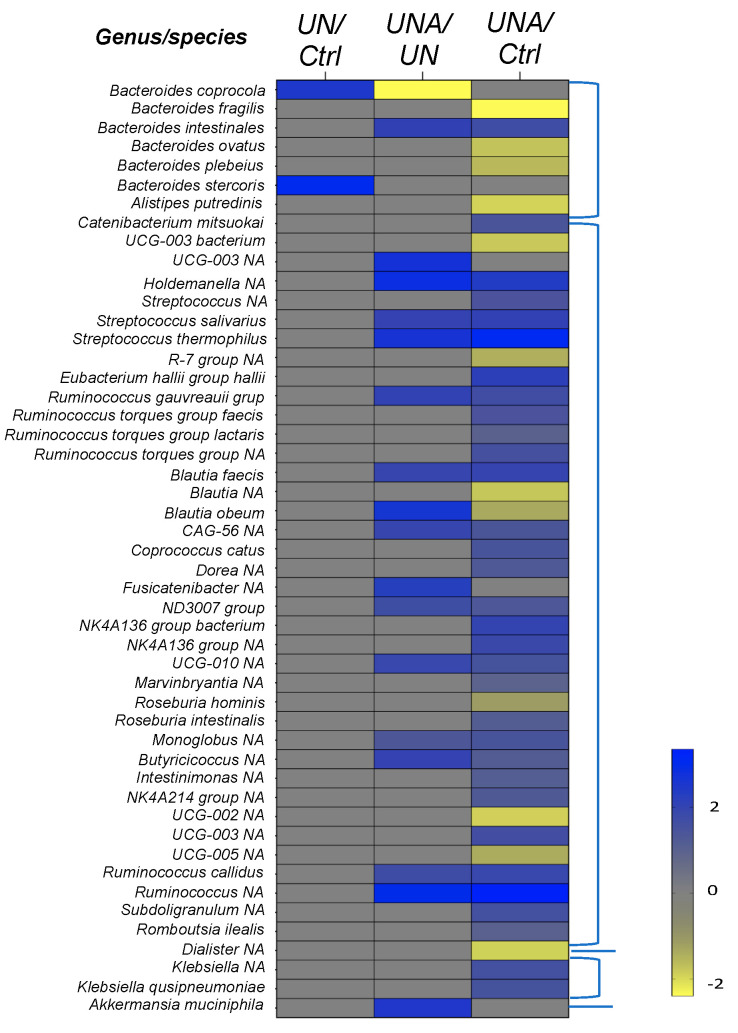
Heatmap of differential ASVs observed between: UN/Ctrl, undernutrition group (U/N) vs. control group (Ctrl); UNA/UN, undernutrition group after amaranth consumption (UNA) vs. undernutrition group at the beginning of assay (UN); UNA/Ctrl, undernutrition group after amaranth consumption (UNA) vs. control group (Ctrl). Values were determined using LogFC, at *p* < 0.05 and FDR < 0.1.

**Figure 6 foods-12-02760-f006:**
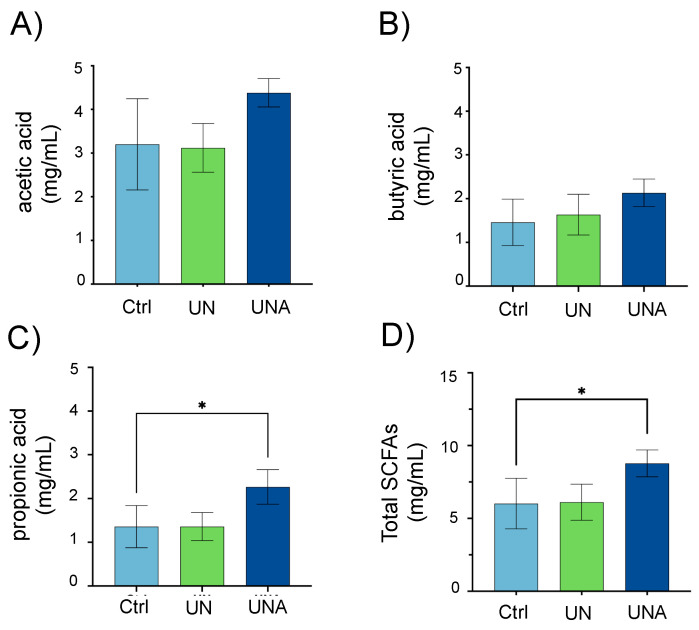
Short-chain fatty acid (SCFA) levels in children’s feces samples. One-way ANOVA was carried out followed by Tukey’s post hoc test. (**A**) acetic acid; (**B**) butyric acid; (**C**) propionic acid; (**D**) Total SCFA. Data is presented as mean of mg/mL ± standard deviation. Asterisk shows significant differences at *p* < 0.05. Ctrl = control group of children normal weight-for-age; UN = undernutrition children group with low height-for-age; UNA = undernutrition children group after three months of amaranth consumption.

**Figure 7 foods-12-02760-f007:**
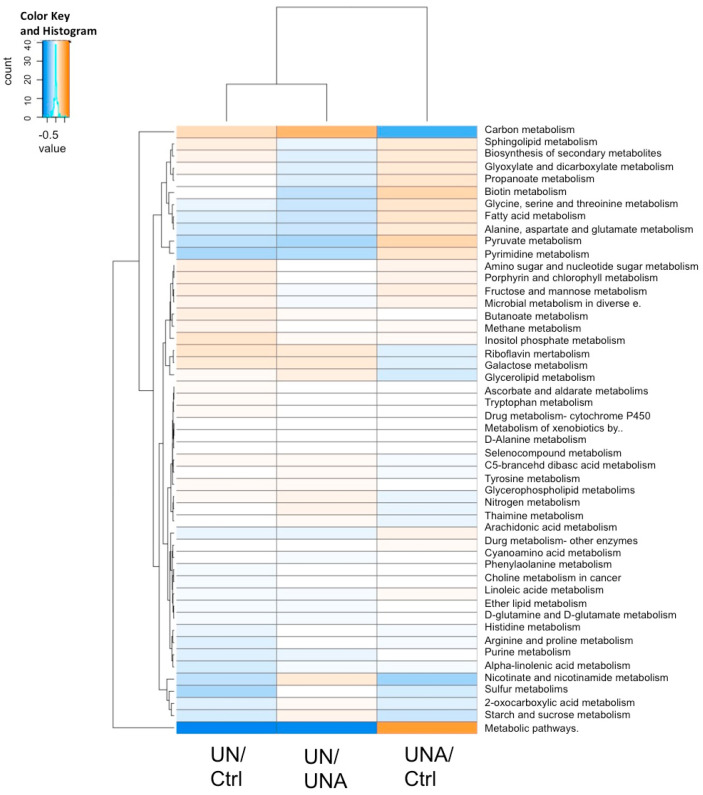
Heatmap of predicted functions of the gut microbiome by Tax4fun2 evaluated according to the differences in the Kyoto Encyclopedia of Genes and Genomes (KEGG) pathway. Differences in the KEGG pathway: undernutrition group (UN) vs. control children (Ctrl); undernutrition group after amaranth consumption (UNA) vs. undernutrition group at the beginning of assay (UN); undernutrition group before amaranth consumption (UN) vs. control group (Ctrl).

**Figure 8 foods-12-02760-f008:**
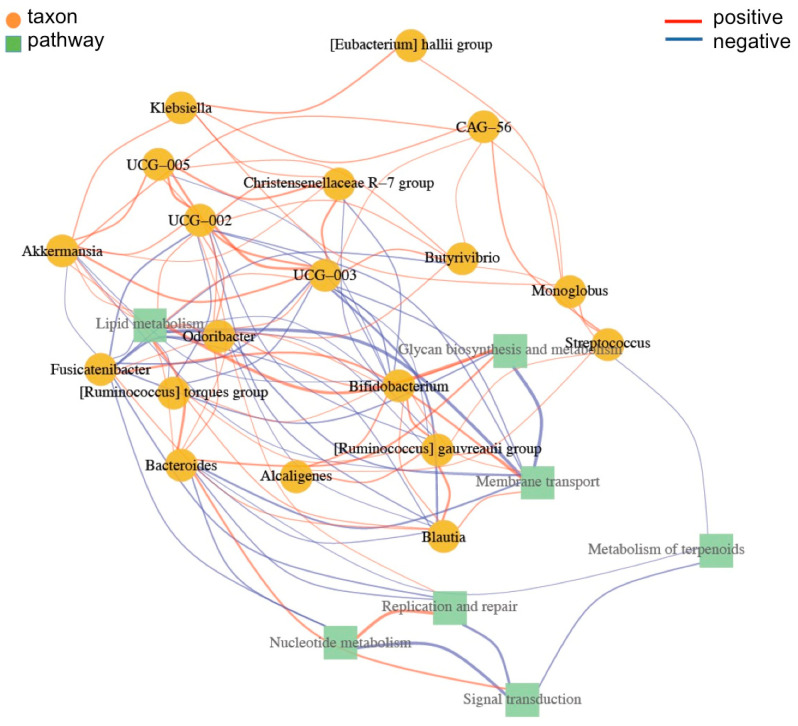
Association between taxa and pathways. Taxa are represented by yellow circles and pathways by green squares. The network’s nodes are taxa and pathways. The association between the nodes and edges was calculated using the Spearman correlation methodology. Only correlations of absolute co-ordinates > 0.5 were considered. The nodes correlations are represented by the vertex, where red are positive and blue negative correlations.

## Data Availability

The data presented in this study are available on request from the corresponding author. The data are not publicly available due is in process to uploaded into public database.

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
