# Peer review of "Dietary Supplementation with Popped Amaranth Modulates the Gut Microbiota in Low Height-for-Age Children: A Nonrandomized Pilot Trial"

_foods, 2023, doi:10.3390/foods12142760_

Round 1

Reviewer 1 Report

Malnutrition in children is a major social problem. The authors have undertaken an interesting research task. The cited References are relevant to the topic of the paper. Interesting results for a pilot study.

Please, add in the introduction information on whether there are reports on the effect of other plant raw materials on the intestinal microbiota of children with malnutrition and motivate why amaranth was chosen.

Fig. 3 not very readable

p.26 of the literature - to be checked

Author Response

point-by-point answers:

Point 1. Malnutrition in children is a major social problem. The authors have undertaken an interesting research task. The cited References are relevant to the topic of the paper. Interesting results for a pilot study.

Answer: thanks for your comments

Point 2. Please, add in the introduction information on whether there are reports on the effect of other plant raw materials on the intestinal microbiota of children with malnutrition and motivate why amaranth was chosen.

Answer: information was added and marked in yellow

Point 3: p.26 of the literature - to be checked

Answer: literature was reviewed.

Reviewer 2 Report

The authors aimed to explore the effect of popped amaranth consumption (4g/daily per 3 months) on the changes in the structure and abundance of gut microbiota of children classified as low height for age. Currently, there is no information on the impact that popped amaranth consumption has on gut microbiota composition in children. Hence the authors bring new information regarding this topic in their original pilot study. Currently, there is no information on the impact that popped amaranth consumption has on gut microbiota composition in children. Hence the authors bring new information regarding this topic in their original pilot study. I believe that the study design is appropriate and there is no need to change the methodology. The conclusion of the evaluated paper is consistent a sufficient. It is complex and summarizes the result of the study. The manuscript reviews 74 articles representing the sum of the current knowledge regarding the topic giving appropriate background information. In the evaluated paper authors used 8 figures and no table. The amount of figures is sufficient aid to the reader's understanding. I suggest checking for some spelling mistakes and grammar errors.

I suggest checking for some spelling mistakes and grammar errors.

Author Response

Point 1: I suggest checking for some spelling mistakes and grammar errors.

Answer: Thanks for your comments, manuscript was checked and corrected the grammar errors.

Reviewer 3 Report

This study investigates the effect and biochemical mechanism of Dietary supplementation with popped amaranth modulates the gut microbiota in low height-for-age Children. The topic is interesting. However, there are some problems in this study. For example, the background about the relationship between malnutrition and gut microbiota dysbiosis is unclear. Some important information is not surveyed, including the causes of low weight-for-age, food composition. Moreover, some methods are not introduced. In addition, the discussion need to be further improved. Thus, I recommend that the manuscript needs a major revision.

After reviewing a paper, I suggest the following corrections:

1. In the part of “Introduction”: the author referred that Persistent childhood malnutrition is considered part of a vicious cycle of recurrent infections, impaired immunity, and worsening malnutrition, and all these factors have been linked to gut microbiota dysbiosis, which can affect the immune response, the susceptibility to infection, and the nutritional status, resulting in the main consequences of undernutrition, however, the persistent childhood malnutrition is mainly caused by an imbalance in energy intake and energy expenditure. The imbalance in energy intake and energy expenditure can cause gut microbiota dysbiosis. Thus, please confirm the logical relationship between malnutrition and gut microbiota dysbiosis.

2. In the part of “2.1. Recruitment of participants”: When recruiting the participants, in addition to the body weight, it is necessary to survey the causes of low weight-for-age, which is very important for this study. Because different reason can affect the gut microbiota.

3. In the part of “2.2. Research design”: It is necessary to provide the food composition in addition to popped amaranth, which can disturb the gut microbiota. Moreover, whether these participants are sick is also very important during the study.

4. In the part of “2.4. Research outcome ”: When and how to obtain the sera and stool sample? In the morning, afternoon, or evening?Before or after eating food?

5. In the part of “2.5. Serum collection and biochemical profile analysis ”: Why these indicators are determined? What’s the relationship between these indicators and gut microbiota?

6. In the part of “3.2. Children´ s selection and serum biochemical analysis ”: Although the hematic cytometry of participating children are described in this part, the corresponding determination method is not mentioned in the part of “Materials and Methods”.

7. Pg.9: The reference of 39 is inappropriate used in this part, because this is about quinoa protein not amaranth.

8. Pg.12: The reference of 45 is inappropriate used in this part.

9.  It is important to combine the discussion between gut microbiota and gut microbiota-dependent metabolites, because the SCFAs are produced by gut microbiota and the variation of gut microbiota can affect the SCFAs content.

10. Please simplify the conclusion.

11. In the part of “References”: Please unify the DIO format.

12. Microorganism names in Latin italics.

13. Figure 2 Error lines are not fully displayed and p&lt has been marked; 0.05, but the analysis is reflected.

14. Figure 3 The picture definition is low, it is recommended to replace the vector picture.

15. Figure 8 lacks analysis.

Author Response

point-by-point answers:

After reviewing a paper, I suggest the following corrections:

point 1. In the part of “Introduction”: the author referred that Persistent childhood malnutrition is considered part of a vicious cycle of recurrent infections, impaired immunity, and worsening malnutrition, and all these factors have been linked to gut microbiota dysbiosis, which can affect the immune response, the susceptibility to infection, and the nutritional status, resulting in the main consequences of undernutrition, however, the persistent childhood malnutrition is mainly caused by an imbalance in energy intake and energy expenditure. The imbalance in energy intake and energy expenditure can cause gut microbiota dysbiosis. Thus, please confirm the logical relationship between malnutrition and gut microbiota dysbiosis

Answer: thanks for your observation, this part of introduction was modify and simplify.

point 2. In the part of “2.1. Recruitment of participants”: When recruiting the participants, in addition to the body weight, it is necessary to survey the causes of low weight-for-age, which is very important for this study. Because different reason can affect the gut microbiota.

Answer: data of recruitment was added. As you said, malnutrition is a very complex with many factors involved more than only diet. Yes, we did a sociodemographic survey, now is added .

point 3. In the part of “2.2. Research design”: It is necessary to provide the food composition in addition to popped amaranth, which can disturb the gut microbiota. Moreover, whether these participants are sick is also very important during the study.

Answer:  means what type of food are normally the children eat ?, this info was added. As we mentioned in section 2.1, recruitment of participants. exclusion criteria was sick children.

point 4. In the part of “2.4. Research outcome ”: When and how to obtain the sera and stool sample? In the morning, afternoon, or evening?Before or after eating food?

Answer: this info was added in section 2.5 serum collection.

blood samples were taken in the morning under fasting conditions...

point 5. In the part of “2.5. Serum collection and biochemical profile analysis ”: Why these indicators are determined? What’s the relationship between these indicators and gut microbiota?

Answer: serum biochemical analysis were used as a parameter to identify if children had any disease more other than to be low height-for-age. The paramaters measured should not have the intention to related with gut microbiota structure and function more than the health status of children.

point 6. In the part of “3.2. Children´ s selection and serum biochemical analysis ”: Although the hematic cytometry of participating children are described in this part, the corresponding determination method is not mentioned in the part of “Materials and Methods”.

Answer: Thanks for your observation. determination of cytometry analysis has been added in section 2.5.

point 7. Pg.9: The reference of 39 is inappropriate used in this part, because this is about quinoa protein not amaranth.

Answer: as recommended by you and other reviewer, this reference was moved to introduction.

point 8. Pg.12: The reference of 45 is inappropriate used in this part.

Answer: Thanks , reference was changed

point 9. It is important to combine the discussion between gut microbiota and gut microbiota-dependent metabolites, because the SCFAs are produced by gut microbiota and the variation of gut microbiota can affect the SCFAs content.

Answer:  in discussion section it is described the bacterium and it is indicated how it was classified as acetate-, propianate- or butyrate-producer bacterium. In Section 3.6 we were focus on quantification of SCFA, as general observation of changes in abundance of bacteria. 

point 10. Please simplify the conclusion.

Answer: conclusion was simplify.

point 11. In the part of “References”: Please unify the DIO format.

Answer: DOI was unify

point 12. Microorganism names in Latin italics

Answer: thanks, all genera-species names are written in italics, phylum and families are not.

point 13. Figure 2 Error lines are not fully displayed and p&lt has been marked; 0.05, but the analysis is reflected.

Answer: error lines were aded as ticker black lines

point 14. Figure 3 The picture definition is low, it is recommended to replace the vector picture.

Answer: figure 3 was changed with a better resolution

point 15. Figure 8 lacks analysis.

Answer: figure legend was corrected adding the info about analysis.

Round 2

Reviewer 3 Report

The manuscript has been sufficiently improved to warrant publication in Foods.

Author Response

The manuscript has been sufficiently improved to warrant publication in Foods.

answer:

thanks for your questions and manuscript review